# Current and Potential Roles in Sports Pharmacy: A Systematic Review

**DOI:** 10.3390/pharmacy7010029

**Published:** 2019-03-14

**Authors:** Alison D. Hooper, Joyce M. Cooper, Jennifer Schneider, Therése Kairuz

**Affiliations:** 1Faculty of Health and Medicine, School of Biomedical Sciences and Pharmacy, University of Newcastle, Callaghan 2308, Australia; Joyce.Cooper@newcastle.edu.au (J.M.C.); Therese.Kairuz@newcastle.edu.au (T.K.); 2Faculty of Health and Medicine, School of Medicine and Public Health, University of Newcastle, Callaghan 2308, Australia; Jennifer.Schneider@newcastle.edu.au

**Keywords:** pharmacy, doping, performance-enhancing, medicine, supplement, sport, athlete, injury

## Abstract

(1) **Background**: The objective of this systematic review was to evaluate current and potential roles for pharmacists in sports medicine and to identify key themes in outcomes reported in studies. (2) **Methods**: EMBASE, MEDLINE, CINAHL, Scopus and the Cochrane Library were searched in January 2019. Peer-reviewed, original research articles were considered for inclusion. Articles published in a language other than English were excluded. Quality appraisal was performed independently by two authors. (3) **Results**: Findings of 11 eligible articles (10 observational and 1 experimental study design) were grouped into three themes: (i) doping prevention and control, (ii) injury management and first aid, and (iii) educational and curricular needs. Pharmacists were perceived as a good potential source of information about doping and are enthusiastic about counseling athletes, but lack knowledge and confidence in this area. While pharmacists were frequently consulted for advice on managing sprains and strains, their advice was not always guided by current evidence. Pharmacists and pharmacy students recalled limited opportunity for education in sports pharmacy. (4) **Conclusion**: Pharmacists showed a willingness and an aptitude to counsel athletes. However, lack of knowledge and confidence, and limited educational opportunities, were key barriers. More research is necessary to support pharmacists in this role.

## 1. Introduction

The role of the pharmacist continues to evolve and develop. Pharmacist-delivered influenza vaccination programs and prescribing are examples of how the scope of pharmacy practice is expanding worldwide [1,2,3,4]. To a lesser extent, roles for pharmacists in various aspects of sporting culture and competition have a limited description in the literature, and range from providing advice for non-competitive/individual fitness and local club sports, to advice for elite athletes competing in the Olympic Games or in other international arenas [5,6,7,8,9]. Roles have been described in doping and anti-doping, injury management and prevention, and first aid [6]. While several authors have produced guidelines for pharmacists regarding the management of common sports injuries and providing counseling/advice to athletes [10,11,12,13,14,15,16], the guidelines do not draw on original research, and a systematic review on sports pharmacy has not been previously performed. Although this emerging field offers new opportunities for pharmacists, without adequate support, the additional responsibility may become burdensome.

In 2005, the Statement of Professional Standards: “The Role of the Pharmacist in the Fight Against Doping in Sport” was adopted by the International Pharmaceutical Federation (FIP), and transformed into FIP Guidelines in 2014 [17]. The guidelines refer to the World Anti-Doping Agency (WADA) Anti-Doping Code [18] and provide recommendations around doping control relevant to pharmacists, governments, and pharmaceutical associations. Recommendations for pharmacists include: (i) keeping up-to-date with the contents of the WADA Code, (ii) assisting athletes to recognize whether the use of a substance may be banned or restricted in their sport, and (iii) providing information to athletes about the risks and benefits of nutritional supplements; many supplements contain banned substances according to the WADA Code or specific sports-governing parties [17,19]. Pharmacists with expertise in sports medicine must stay current with new and changing recommendations concerning the use of appropriate products, and this study reviews educational opportunities for pharmacists in sports pharmacy.

Provision of drug-information and education services to athletes, coaches and supporters may also fall within the scope of “sports pharmacy”. A pharmacist with expertise in sports pharmacy could assist in the identification, management, and monitoring of athletes who seek performance-enhancing supplements and who may, otherwise, experience adverse effects. In addition, assisting in the management of medical conditions such as exercise-induced asthma is another area in which pharmacists could utilize their expertise to support athletes [20].

Doping “prevention and control” encompasses drug-use for therapeutics and performance enhancement, and pharmacist involvement requires knowledge and interpretation of banned substance lists, advising on dietary supplements, over-the-counter (OTC) and prescription medicines and products, tailored formulary development, inventory control, and record keeping [6]. Pharmacists have also been involved in collecting specimens for drug-testing [20].

Common sports injuries such as sprains and strains may cause long-term pain and functional deficiency [21]. While a sprain is defined as a stretched or torn ligament, a strain refers to the overstretching or tearing of a muscle or tendon. Although fundamentally different injuries, both elicit an inflammatory response and the general principles of treatment are the same [10]. Sprains and strains were the most common type of musculoskeletal injury managed by Australian general practitioners between April 2000 and March 2015. Around 37 musculoskeletal injuries were managed for every 1000 consultations, and about 40% of these injuries were sprains or strains [22]. To our knowledge, there is no data available to describe the frequency of presentation of soft tissue injuries presenting to community pharmacies as a primary care setting. Nonetheless, pharmacists may play an important role in advising on treatment, prevention, and referral for sprains and strains, particularly given the belief that this type of injury is usually minor and self-treatable [23]. Evidence also suggests that patients often take OTC products inappropriately due to the perceived safety of these agents [24]. There is a growing body of evidence indicating that the use of anti-inflammatory drugs in the acute inflammatory phase (during the first 24–48 h post-injury) may be detrimental [10,25]. Pharmacist intervention could potentially reduce the risk of further tissue damage by conducting early assessment and providing appropriate treatment recommendations [10].

The underpinning research question for this study was “What are the current and potential roles for pharmacists in sports medicine”? This systematic review aims to examine the literature to identify sports-related health needs that may benefit from pharmacist expertise, as well as the attitudes, beliefs, and knowledge of pharmacists, athletes, and other key stakeholders concerning the roles pharmacists can play in caring for the athlete. For the purpose of this article, the term “sports pharmacy” refers to the provision of pharmacy services in sport. The “athlete” is a person who pursues an individual or team sporting/fitness venture at any level.

## 2. Methods

The PRISMA (Preferred Reporting Items for Systematic reviews and Meta-Analyses) guidelines were consulted in conducting this systematic review [26].

### 2.1. Search Strategy and Study Eligibility

The following electronic databases were searched systematically: EMBASE, MEDLINE, CINAHL, Scopus, and the Cochrane Library. Search terms are outlined in Table 1.

Hand-searching of the reference lists of published studies was conducted to identify additional relevant studies. Articles considered for inclusion were original research published in peer-reviewed journals. No limits were applied to the publication date or geographical location. Articles were excluded if they were published in a language other than English (*n* = 2). Searches were performed by the first author (AH) with the guidance of a senior librarian. Of the articles obtained from the initial search, titles were screened for potential eligibility. Due to the limited amount of published literature on sports pharmacy, any title that appeared to involve pharmacists or pharmacy students practicing in any pharmacy setting, and within any area of sports-related health, was considered for inclusion. Grey literature was searched using Google™. Full-text review was undertaken. Shortlisted articles (*n* = 11) were subjected to quality appraisal.

### 2.2. Data Extraction and Quality Appraisal

The primary outcomes of interest were “any” role undertaken by pharmacists or pharmacy students in sports-related health, as well as outcomes that described knowledge, attitudes, beliefs, or practices of pharmacists or key stakeholders in pharmacist-provided healthcare/services in sports medicine. In addition to outcomes, the following data were extracted from eligible papers: publication year, author details, study location and year, sample size, and study design, context, and methodology.

Quality appraisal was performed using appropriate checklists from the University of Oxford Center for Evidence-Based Management (CEBM, Critical Appraisal of a Survey) [27] or The Joanna Briggs Institute (Checklist for Quasi-Experimental Studies, Non-randomized Experimental Studies) [28]. The CEBM is an independent authority on evidence-based practice. The Critical Appraisal of a Survey tool is used to assess a study for appropriateness of the study design, sample selection and size, response to and validity of questionnaires, and significance and applicability of results. The Joanna Briggs Institute administers critical appraisal tools developed by the research and development Center within the Faculty of Health and Medical Sciences at the University of Adelaide, South Australia. The Checklist for Quasi-Experimental Studies guides the reviewer in assessing articles for a clear “cause and effect”, participant/group variables, outcome measures, follow-up, and statistical analysis. Responses to appraisal questions were allocated a score of “2”, “1” or “0” for responses of “yes”, “can’t tell”/ ”unsure”/ ”not applicable” or “no” respectively. Each eligible article was subject to independent quality appraisal by the first author and a co-author (either TK or JC). Studies were classified as high, moderate, or low quality if they scored the positive response for at least 75%, 50%–74%, or less than 50%, respectively. Discrepancies in scores were discussed until consensus was reached. After removal of duplicate records, the search strategy identified 104 potentially eligible articles. An initial screening of titles reduced the number of potential publications to 16. Two articles were excluded after abstract screening, and, after full-text screening, there were 11 articles deemed eligible for inclusion. Reasons for ineligibility were studies published in a language other than English and published conference abstracts.

## 3. Results

This systematic review identified current roles for pharmacists in sports pharmacy. Findings of 11 eligible articles were grouped into three themes: (i) pharmacist involvement in doping prevention and control, (ii) injury management and first aid, and (iii) educational and curricular needs of pharmacists and pharmacy students in sports pharmacy.

The search strategy and process of article selection are illustrated in Figure 1. Studies were published between 2000 and 2018, and common outcomes included knowledge, attitudes, perceptions, and skills of pharmacists concerning the provision of pharmacy services to athletes, experiences of/roles played by pharmacists or pharmacy students concerning doping prevention and control, injury management and first-aid, attitudes of athletes toward pharmacists as a source of information about drugs in sport, and educational/curricular needs of pharmacists in sports pharmacy. Studies were conducted in the United States of America (USA) (*n* = 3), Qatar (*n* = 2), Canada (*n* = 2), Iran (*n* = 1), New Zealand (*n* = 1), Malaysia (*n* = 1), and France (*n* = 1).

### Quality Appraisal, Study Characteristics, and Summary of Results

Characteristics of the 11 eligible studies are summarized in Table 2. Study design of most (81.8%) of the studies was “observational” [19,20,23,29,30,31,32,33,34,35] with only one “experimental” design [36]. Although observational cross-sectional studies have a place in research, particularly in emerging fields such as sports pharmacy, most (*n* = 7/11) used convenience sampling [19,20,29,31,33,35,36] and all utilized surveys, which were piloted only for face validity or content validity. This may be due to a lack of funding since only one of the studies was supported by an external granting agency [32]. Despite this limitation, the observational studies by Mottram et al., Braund et al., and Chiang et al. followed rigorous design, data collection, and analysis methods. All studies had received ethical approval.

There was a large variability in response rates (ranging from 20.6% to 92.7%, mean response rate 65.0%). Two studies involved an evaluation of pharmacy student experiences of university-administered programs: An Advanced Pharmacy Practice Experience (APPE) in doping control (descriptive educational study) [20] and provision of first aid at sporting events (pre-survey and post-survey of participants) [36].

Studies involving surveys or questionnaires concerning pharmacists’ roles in doping control and prevention analyzed responses from at least one of the following groups: pharmacists, athletes, and/or coaches. Key outcomes were knowledge, attitudes, experiences, practice, and influences in relation to pharmacists’ roles concerning drugs and supplements in sports, as well as medication use in athletes, sources of drug/supplement information for athletes, and educational/curricular needs of pharmacists. A study in Canadian athletes competing at the International Powerlifting Federation Classic World Championships reported findings from a pharmacist-administered assessment of medication and supplements among athletes. The key finding was that 48% of competitors were inadvertently taking at least one substance listed on the WADA Prohibited List, which was most commonly pseudoephedrine [35].

A study conducted in New Zealand analyzed pharmacists’ responses to survey questions about the frequency of the presentation of sprains and strains, the types of advice provided by pharmacists, and their beliefs about the role of analgesics in managing these injuries [23]. The authors found that, while pharmacists are frequently consulted for advice on managing sprains and strains, their advice was not always guided by current evidence questioning the benefits of non-steroidal anti-inflammatory drugs (NSAIDs) in the initial 24 to 48 h after injury.

## 4. Discussion

“Sports Pharmacy” is an emerging specialty within pharmacy practice. We have explored roles that are currently filled by pharmacists in this domain and have exposed the potential for expansion of current practice, particularly in areas of doping prevention and control, as well as injury prevention and management, and educational opportunities. Notably, the lack of educational opportunities for pharmacists and pharmacy students to learn and practice aspects of sports pharmacy places significant burden on pharmacists wishing to provide evidence-based advice to athletes. Pharmacists have the potential to contribute to positive health outcomes for athletes at all levels of recreation and competition, and benefits of pharmacist involvement could extend to providing sports-related medicine and drug advice to coaches and managers, clubs and organizations, family members, and supporters.

The FIP Statement on “The Role of the Pharmacist in the Fight against Doping in Sport” demonstrates an identified and widespread need for pharmacists to play a pivotal role in sports medicine [17]. Of the 104 records identified through this systematic search, only 11 relevant articles describing original research were eligible for inclusion. This is suggestive of a significant lack of published literature exploring current and potential roles for pharmacists in sports medicine. Pharmacy practitioners are unlikely to be able to draw on quality research to inform practice in this field, which adds to the burden on pharmacists who offer consulting services to athletes in practice. Our findings also highlight a need for funding from local, national, or government organizations to contribute to a drug-related societal issue that is of interest and concern to all countries that compete in sporting events.

### 4.1. Doping Prevention and Control

In partnership with the Japan Pharmaceutical Association, the Japan Anti-Doping Agency (JADA) established an initiative in 2009 that integrates expertise from a range of health professions to promote anti-doping [37]. As part of the program titled “Play True 2020”, pharmacists undertake a certification process incorporating basic and practical components to become certified as Sports Pharmacists by JADA, who are then considered professionals with the most up-to-date information on anti-doping guidelines. Sports Pharmacists aim to provide health education and advice on the safe and rational use of substances to athletes. In 2014, there were 6000 certified Sports Pharmacists in Japan. Research supporting this program was unable to be comprehensively evaluated as part of this systematic review due to language constraints.

Pharmacists consider themselves a good potential source of drug information for athletes [19,31,32], and, while pharmacists are enthusiastic about counseling athletes, few feel equipped to do so [29,31,32,34]. Key barriers identified by pharmacists were a lack of knowledge about prohibited lists, not knowing where to source reliable information about banned substances, and a lack of confidence in discussing doping with athletes [19,36]. Other barriers included time, a perceived lack of evidence about the benefits of sports supplements, and a limited range of sports supplement products stocked by pharmacy outlets [19,31].

Athletes considered pharmacists to be a good potential source of information even though they did not often consult a pharmacist when purchasing drugs and supplements [19,33]. On the other hand, pharmacists felt that athletes usually purchase supplements online rather than consulting health professionals [31], which is a belief that was supported by the findings of Howard et al. [19]. While pharmacists were consulted by athletes between 2% and 35% of the time [19,33], athletes who completed the surveys indicated that they would consider seeking advice from a pharmacist more often since they were considered trustworthy [19,33]. However, athletes do not consider advertisements, endorsements, and websites trustworthy [19]. Other sources of information for athletes about doping included other players, coaches, team managers, and the Internet, even though athletes generally rated information from these sources as less reliable when compared to consulting a health professional [29,34].

Pharmacists have demonstrated that they have the skills and knowledge to elicit a thorough medication history from athletes [35,36]. However, an evaluation pharmacists’ ability to identify prohibited substances during pharmacist-client interviews has not been undertaken. Similarly, pharmacist interpretation of the WADA code and other guidelines administered by local sports-governing agencies has not been directly evaluated. While doping was considered a public health problem by most pharmacists [32], the majority in one study in Qatar were unaware of WADA’s role and even fewer had an awareness of the FIP Statement [29]. Additionally, in a similar study in Qatar, most pharmacists were aware of WADA but not the FIP Statement [34]. Of concern was the fact that nearly one-fifth of pharmacists were unaware of a list of banned substances that was kept at their place of practice [32].

#### 4.1.1. Pharmacist’s Knowledge

Pharmacists rated their knowledge and awareness of doping and anti-doping as low [30,34], even though they are interested in receiving education about doping [29]. Articles about doping in industry publications were a useful source of information for some pharmacists [31,32]. Pharmacists with no specific training or qualifications were unable to correctly classify most substances as prohibited or permitted in sports. In the included studies, cough and cold products and narcotics were the only substances that were usually correctly identified by pharmacists as banned substances. Pharmacists demonstrated a varying ability to identify anabolic steroids, cannabis, other stimulants, insulin, and dietary and ergogenic substances [29,32,34]. Substances most commonly taken by athletes in this systematic review were pseudoephedrine, protein, vitamins, weight gain powders, amino acids, creatine, and caffeine [19,31,35].

#### 4.1.2. Other Potential Roles for Pharmacists

There is potential for pharmacists to participate in drug-testing and doping control. Pharmacists and pharmacy students have demonstrated the ability to competently conduct specimen collection for drug-testing purposes and provide education to colleagues about drugs in sports [20]. Roles in provision of conventional pharmacy and anti-doping services at the Olympic Games have also been reported in the literature [7,8,9]. At the London 2012 Games, more than 100 volunteer pharmacists and pharmacy technicians were stationed at polyclinics within the Athlete Villages and pharmacy services were provided in accordance with UK Standards [7]. The supply, management, and storage of medicines for use at training and competition venues was facilitated by pharmacists through these ‘polyclinic pharmacies,’ guided by a specialized set of policies and standard operating procedures. Pharmacists were responsible for writing and reviewing Olympic and Paralympic drug formularies and delivering an international drug information service to prescribers at the Games. A Pharmacy Minor Ailments Scheme was also implemented and pharmacists provided pharmacy services to non-athletes to relieve the workload on fellow health professionals at the Games polyclinics. In consultation with the Center for Pharmacy Postgraduate Education (CPPE) a 12-week online e-learning program entitled ‘The Use of Drugs in Sport: A Healthcare Professional’s Perspective’ was developed specifically for the 2012 Olympic Games. The program was divided into three sections: doping and anti-doping in sport, pharmacy services and support in sport and fitness, and medical services at international sporting events [7]. The education was a pre-requisite for pharmacy volunteers and was also made available online via the CPPE website to all pharmacists, pharmacy technicians, and health professionals in the UK (Program no longer available) [38]. A similar program, on a much smaller scale, was introduced for the Commonwealth Games at the Gold Coast in Australia, in 2018 [39].

### 4.2. Injury Management and First Aid

Australian community pharmacies typically stock a variety of pharmaceutical and non-pharmaceutical products available over-the-counter, which are marketed for various sporting and soft tissue injuries. Products include a range of straps and supports, mobility aids, supplements, and analgesic and anti-inflammatory products. Studies involving Australian pharmacists and sports pharmacy have not been published, although results of a New-Zealand study suggest that pharmacists provide advice that is conflicting and not always in line with current evidence-based practice, particularly regarding NSAID use immediately post-injury [23]. This occurs despite pharmacists frequently being confronted with sprains or strains. Regarding first aid, pharmacy students were able to recognize ‘red flags’ in injured or dehydrated athletes [36] and pharmacists could triage patients when considering referral to hospital or another health care professional [23,36]. More research is needed to support pharmacists in making evidence-based recommendations for athletes, and targeted education for pharmacists could be of value to pharmacists and consumers.

### 4.3. Opportunities for Education and Training and Recommendations Arising from This Review

This systematic review has discussed gaps in pharmacist knowledge and awareness about the prohibited status of drugs. Despite an enthusiasm for counseling athletes about drugs in sport, pharmacists in the included studies lacked confidence in counseling athletes about licit and illicit drugs [19]. Gaps, and lack of confidence, were key barriers to service provision. Increasing opportunities for pharmacists to receive education and to practice skills in sports pharmacy may increase pharmacist knowledge and confidence and overcome these barriers, which was demonstrated by Ambrose (2008) and Doty et al. (2015) [20,36].

Pharmacists reported limited opportunity for education in sports pharmacy in both undergraduate curricula and postgraduate education. While research has not evaluated the extent of inclusion of sports pharmacy in undergraduate curricula, only 6% of pharmacists in one study [32] and 26% in another study [19] recall being taught about doping control at a university level. Literature has few university-based experiences in sports pharmacy [20,36,40]. A study among pharmacy students in Qatar reported that most (90%) were in favor of incorporating sports pharmacy into undergraduate curricula [29]. Self-interest was the driver of knowledge acquisition for more than one-quarter (28%) of pharmacists [19]. Pharmacists were most interested in short post-graduate/professional development courses delivered either online or face-to-face [19,34] even though few pharmacists had attended postgraduate courses in sports pharmacy, which may have been due to limited opportunity rather than a lack of interest [31].

Sprains and strains are a common presentation to community pharmacies, and pharmacists would benefit from educational opportunities on the evidence-based management of soft tissue injuries. Braund et al. reported that around half of pharmacists in a study in 2005 recommended NSAIDs to patients in the first 24 to 48 h after a sprain or strain despite evidence suggesting benefits of withholding NSAIDs during this initial period [41]. Pharmacists cited reasons including a belief that inflammation is a barrier to healing, that the patient is entitled to analgesia or that there is no reason to withhold. While more research is needed into the types of recommendations, pharmacists make for acute soft tissue injury, this study demonstrates a need for educational opportunities for pharmacists in soft tissue injury management.

### 4.4. Limitations

This systematic review is not without limitations. A notable shortfall was the exclusion of articles published in languages other than English, which may have contributed to the lack of information from Asian, African, and South-American countries. This is particularly relevant for two articles published in Japanese given the initiatives of the Japan Anti-Doping Agency. Furthermore, only 11 articles were considered eligible for review.

## 5. Conclusions

Pharmacists have the willingness and expertise to help address the public health issue that is doping in sports, and which has been recognized in Japan through JADA’s Sports Pharmacist certification program. Although pharmacists appear to be enthusiastic about counseling athletes, lack of knowledge and confidence in sports pharmacy is a key barrier to their involvement. Opportunities for pharmacists to obtain knowledge and practical experience in sports pharmacy are necessary in both undergraduate and postgraduate pharmacy programs. More research is necessary to support pharmacists in this role. For example, government and industry support is needed to fund more research into current practices and potential roles for pharmacists in sports medicine.

## Figures and Tables

**Figure 1 pharmacy-07-00029-f001:**
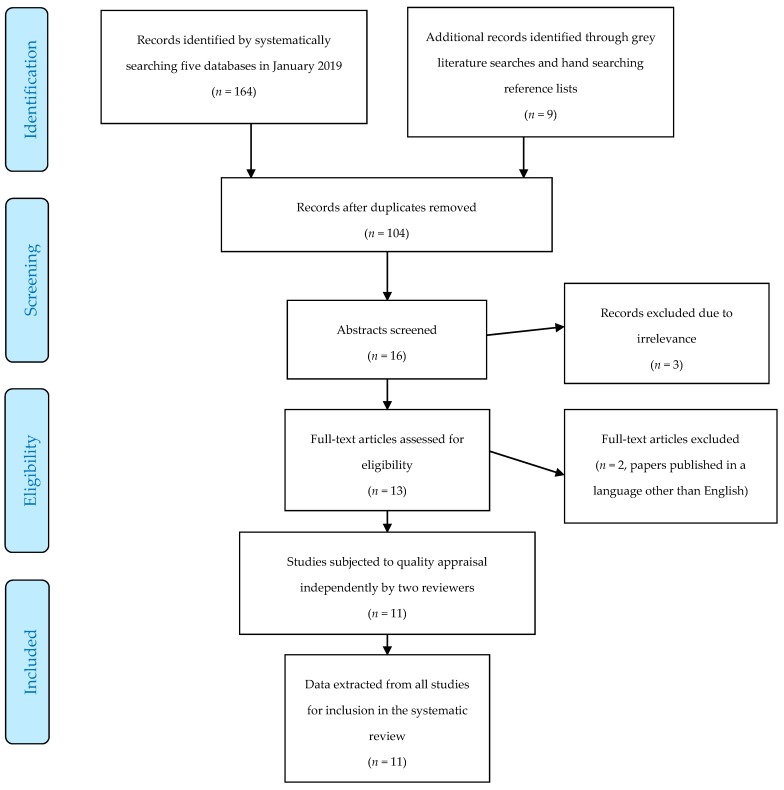
Search strategy and process of article selection.

**Table 1 pharmacy-07-00029-t001:** Search terms used in the literature retrieval.

**Pharmacist* OR Pharmacy OR Pharmacies**
AND
drug* OR ailment OR medicin* OR substance OR medical or test* OR sample or urine or urinalysis OR advise OR advice OR dispens* OR compound* OR extemporane* OR manufactur* OR inventory OR imprest OR record OR counsel* OR performance OR enhance OR enhancing OR performance-enhancing OR dope OR doping OR anti-doping OR supplement* OR non-steroidal OR anti-inflammator* OR antiinflammator* OR nonsteroidal OR NSAID* OR analgesi* OR paracetamol OR ibuprofen OR diclofenac OR naproxen OR piroxicam OR mefenamic OR topical OR rubefacient* OR CAM OR “complimentary medicine” OR “alternative medicine” OR TENS OR “transcutaneous electric* nerve stimulation” OR ultrasound OR OTC OR “over the counter” OR non-prescription OR nonprescription OR refer* OR recommend* OR RICE* OR non-pharmacological OR nonpharmacological OR lifestyle OR exercise OR assess* OR manage* OR recommend* OR approach OR strap* OR tape OR taping OR brace* OR device* OR crutch* OR “walking stick” OR wheelchair OR “mobility aid” OR orthopaedic OR orthopedic OR “first aid” OR first-aid OR amphetamine* OR stimulant* OR ephedrine OR adrenaline OR ephedra OR caffeine OR anabolic OR steroid* OR “growth hormone” OR erythropoietin OR EPO OR darbepoetin OR androstenedione OR dehydroepiandrosterone OR creatine OR “biological passport” OR vaccinat* OR immunis* OR immuniz*
AND
injury OR sprain* OR strain* OR contusion* OR “soft tissue” OR ligament OR “muscle tear” OR “torn muscle” OR tendon OR ankle OR sport* OR athlete* OR coach* OR Olympic* OR competition OR competitor OR club

**Table 2 pharmacy-07-00029-t002:** Summary of studies included in the systematic review (*n* = 11).

Author (Date)	Study Location; Context; Design; Duration; Sample Size	Main Outcomes	Study Results
Ambrose (2008) [20]	California, USA; Retrospective evaluation of a six-week elective in Advanced Pharmacy Practice Experience (APPE) at two Californian universities among seven senior students.	Assessment of students’ completion and mastery of skills during APPE, and student evaluation of the APPE—satisfaction with the program and confidence in conducting drug-testing and giving drug information presentations.	All students were assessed as satisfactory for all aspects of drug collection procedures. Student evaluations of the APPE indicated a high level of satisfaction. Suggested improvements were more discussion around sports injuries/management, more drug-testing collections, presentations, and exposure to other sports-governing agencies.
Awaisu (2015) [29]	Qatar, 2014; Three-month cross-sectional, web-based survey of 80 female pharmacy students at Qatar University College of Pharmacy	Participants’ knowledge and perception of drugs in sports, and their views on curricular needs of Sports Pharmacy in undergraduate pharmacy programs.	60% were unaware of WADA’s role and 85% were unaware of the FIP Statement on pharmacist’s role in anti-doping. Majority correctly classified some substances as permitted/prohibited: paracetamol (88%), amphetamines (72%), anabolic steroids (65%) antihistamines (61%); most were unable to correctly classify codeine (86%), insulin (85%) and caffeine (56%). Participants rated health professionals as the most reliable source of information. 90% favored incorporating sports pharmacy into undergraduate pharmacy curricula.
Bastani (2017) [30]	Iran, 2015; Five-month cross-sectional survey distributed to randomly selected bodybuilders, coaches, and pharmacists in Shiraz, South Iran. Results from 287 questionnaires included in analysis (189 body builders, 43 coaches, 55 pharmacists).	Comparison of knowledge, attitudes and practice (KAP) about sports supplements among each population, using a 5-point Likert Scale. Questions explored aspects of sports supplements and access to information, therapeutic considerations (efficacy, safety, dosage and administration), drug-supplement and disease-supplement interactions, impact on performance, involvement of health professionals, and accessing or recommending supplements.	In the three populations, mean KAP response was significantly greater than the intermediate score of 3, except pharmacists’ knowledge (not statistically significant). Younger pharmacists with less work experience had a significantly lower level of knowledge about sports supplements. Mean scores of knowledge and practice were significantly higher in the coaches group, while scores reflecting attitude were highest in the bodybuilders group.
Braund (2006) [23]	New Zealand, 2005; One-month cross-sectional, structured, postal questionnaire of 229 pharmacists throughout New Zealand	Frequency of presentation of sprains and strains to community pharmacies, pharmacist interventions and advice provided, and beliefs about the role of analgesics in treatment of these injuries.	An average of nine injuries were reported per month. 96% of pharmacists recommended rest, ice, compression and elevation (RICE) and 89% frequently recommended analgesics. Of these respondents, 46% recommended non-steroidal anti-inflammatory drugs (NSAIDs; oral or topical), paracetamol (36%), and codeine (8%) if required. Almost half (46%) believed that NSAIDs should not be used in the first 24–48 h post injury. Reasons for not withholding NSAIDs included belief that inflammation is a barrier to healing or that the patient is entitled to analgesia. Reasons for withholding were beliefs that inflammation was required for optimal healing, or that NSAIDs may worsen bruising, increase bleeding, or mask signs of further injury. Other advice included avoidance of heat, massage and alcohol for the 24–48 h post-injury, referral to another health professional, and arnica.
Chiang (2018) [31]	Malaysia, 2015; Five-month cross-sectional survey of 108 community pharmacists in Kuala Lumpur. All community pharmacists in Kuala Lumpur with at least one years’ experience in a community setting were invited to participate.	Pharmacists’ knowledge, experience, and perceptions about doping prevention in community pharmacy and factors that might influence doping.	Overall knowledge score regarding drugs in sport was low. 70% were unable to identify the official Malaysian anti-doping agency. 95% of pharmacists knew that anabolic steroids were prohibited in sport. Only 25% were aware of the prohibited status of beta blockers. Most respondents considered doping prevention initiatives important and believed pharmacists can play a role in doping prevention. Only 25% felt equipped to counsel athletes. 50% felt that athletes source performance-enhancing substances over the internet. 10% of respondents could recall requests for information about doping agents and 55% for performance enhancing supplements. Nutritional supplements most commonly sold were whey protein, weight gain powders, and multivitamins. 38% reported dispensing medicines for performance enhancing or body image purposes. Only six pharmacists had attended a course on drugs in sports, while a larger proportion (31%) had read educational materials.
Doty (2015) [36]	USA, 2011; Twelve-month pre-test and post-test evaluating the experiences of 26 and 27 (respectively) pharmacy students from University of Florida’s College of Pharmacy in assisting Alachua County Fire Rescue personnel provide first aid at football games.	Students’ perceptions including self-reported changes in confidence and perceptions of first aid personnel about the student’s presence. Median scores on a Likert Scale were used to compare pre- and post-assessment responses to the survey, and quantitative data was expressed in means to indicate the direction of change.	Significant improvements were seen in the post-survey for all survey criteria. The greatest improvements in student confidence were in relation to communicating and assisting first aid providers and in recognizing a dehydrated patient. Other improvements were in using interview techniques to obtain information from a patient, obtaining accurate accident information from the patient to evaluate risk to the public, documenting accident events, assisting in dispensing OTC ^1^ products, and recognizing heat exhaustion, alcohol intoxication, a patient in distress, or the need to transfer a patient to the hospital.
Howard (2018) [19]	USA, 2017; Two separate cross-sectional surveys delivered over a one-month period to either 129 athletes at Northwestern fitness centers in Ohio, or to 143 pharmacists in chain pharmacies.	Athletes’ interest in pharmacist-delivered advice about sports supplements, including perceptions of pharmacists providing counseling on sports supplements. Pharmacist survey: pharmacists’ knowledge, confidence, and enthusiasm about sports supplement counseling and perceptions on benefits and barriers to providing the service. Pharmacist knowledge, confidence and enthusiasm was measured on a 5-point Likert Scale.	Most athletes obtained supplements from a grocery store or online. Pharmacists were consulted only 2% of the time. 52% indicated that they would consider seeking a pharmacist’s advice. A doctor was consulted 9% of the time, as was a dietician. More commonly accessed sources of information were supplement stores (44%), friends (32%), or other resources (34%, primarily the internet). Information from advertisements, commercials, endorsements, or websites was not considered trustworthy by 71% of athletes. Supplements most commonly used were protein (75%), vitamins (50%), amino acids (25%), creatine (23%) and caffeine (20%). Pharmacists generally disagreed with statements concerning knowledge or confidence in their ability to counsel on sports supplements. Responses about enthusiasm for this type of service were positive—92% of pharmacists believed providing counseling on sports supplements would be beneficial. Perceived barriers were time (22%), lack of evidence (14%), and lack of knowledge (64%). In addition, 31% of pharmacists recalled no education about sports supplements at university, knowledge stemmed from self-interest in 28% of respondents cases, and 26% recalled some university-provided education.
Laure (2000) [32]	France, 1997; Cross-sectional questionnaire administered via scripted telephone interview among 198 French retail (community) pharmacists who reported having been either involved in programs or questioned by athletes about drugs in sports.	Pharmacists’ knowledge about doping in sports, frequency and contexts in which pharmacists are faced with doping, their attitudes toward doping in sport, and the desire of pharmacists to participate in doping control.	Doping was considered a public health problem by 88% of pharmacists. In addition, 25% reported having been confronted with doping in the previous 12 months, either to provide a product or information. Additionally, 6% recalled having been offered proposals (e.g., financial) to supply performance enhancing substances to an organization (e.g., a sporting club). While 91% of pharmacists believe pharmacists can play a role in doping prevention, 74% felt inadequately prepared to do so. Of the eight classes of drugs prohibited in sport in France in 1997, respondents only correctly identified an average of 1.7 ± 1 (most frequently mentioned were stimulants identified by 24.3% of pharmacists, anabolic steroids 20.6%, and narcotics 8.4%). Furthermore, 82.3% reported having a list of banned products in their pharmacy even though 100% had the Vidal dictionary, which contains the list in the opening pages. While only 6% of pharmacists recalled having been taught about doping control during undergraduate training, 91% had ‘seen’ articles and 62% had read some.
Malek (2014) [33]	Canada, 2012; Three-month cross-sectional questionnaire among 307 athletes who were part of an athletic team for the University of Saskatchewan competing in the Canadian Interuniversity Sport program.	Attitudes of athletes about doping, medication use, sources of information about doping, and whether pharmacists play a role as drug-information-providers for athletes.	Athletes did not feel pressured to dope (96.7%) and did not consider it prevalent or necessary (84.5%), or a risk worth taking (95.4%). Furthermore, 82.9% felt that most of their competitors and colleagues do not use performance-enhancing substances. Conversely, 32.9% believed that some high performance student athletes use anabolic steroids. The majority felt they had good knowledge about banned substances. Medication and supplement use was not affected by fear of doping violations with the exception of cough and cold products (avoided by 20% of athletes). An online doping education program administered by the Canadian Center for Ethics in Sport was most frequently accessed (used 74.5% of the time). In addition, 75.6% of athletes considered pharmacists a good source of information about doping, but only 35% reported speaking to a pharmacist when purchasing an OTC ^1^ medicine. Other reliable sources were physicians (accessed by 48.5% of athletes), physiotherapists (42.5%), and other health professionals (30.2%). Additionally, 86.8% of athletes believed they receive adequate information about doping.
Mottram (2016) [34]	Qatar, 2014; Three-month cross-sectional survey of 300 hospital and community pharmacists in Qatar.	Participants’ knowledge and awareness regarding doping and anti-doping, perception of information sources on drug use in sport, and attitudes toward educational needs in sports pharmacy.	While most pharmacists were aware of WADA, most were unaware of the FIP Statement. Respondents achieved an average score (53.2%) concerning knowledge about drugs in sports. Pharmacists scored higher in their knowledge of the prohibited status of OTC ^1^ cough and cold medicines, and dietary and ergogenic supplements, than for miscellaneous substances (e.g., anabolic steroids, amphetamines, insulin, and cannabis). Respondents perceived all potential information sources as being used equally. Their assessment of sources that should be accessed rated healthcare professionals highest when compared with other athletes, coaches, team managers, and the Internet. Sports medicine doctors were considered the most appropriate source of health-professional-delivered information, followed by pharmacists. Pharmacists had limited awareness of doping and anti-doping but most respondents (81.7%) were interested in receiving certified education or training programs in sports pharmacy.
Smith-Morris (2018) [35]	Canada, 2016–2017; Two-year retrospective, comprehensive analysis of medication and supplements performed by a pharmacist on 27 Canadian athletes competing at the International Powerlifting Federation Classic World Championships.	Descriptive data was reported on the findings of the pharmacist-administered assessment.	Athletes reported use of an average of 11 substances including non-prescription and prescription medicines as well as herbal and sports supplements. Overall, 15 substances were identified as being on the WADA Prohibited List or Monitoring Program and 48% of athletes (*n* = 13) were taking at least one WADA banned substance. The most commonly identified banned substance was pseudoephedrine. Most prohibited products were taken “as needed”, two were administered between three and five times a week, and two were taken daily.

^1^ OTC = Over the Counter.

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
