# Peer review of "Current and Potential Roles in Sports Pharmacy: A Systematic Review"

_pharmacy, 2019, doi:10.3390/pharmacy7010029_

Reviewer 1 Report

Dear Authors,

The literature compilation/ survey is having more importance and makes pharmacist aware of sports pharmacy.

A wide range of literature search was performed and surprising to know only 11 relevant articles are describing the original research. Methods used in the study are adequate and the results covered different countries from different continents. 

Please look at sentence 368, where sprains word was repeated instead of strains.

Surprising to know non-availability of information about developing countries like East Asian, African and South American countries which represent more portion of the population.

Author Response

We thank the reviewers for their constructive critique. We have made every effort to address all the comments and comply with suggestions, as described in the attached document.

Reviewer 2 Report

I liked the topic of “sports pharmacy” for this literature review as an emerging specialty.

The manuscript/review is interesting, unique, and detailed. This is a conscientious, detailed, systematic, and organized approach to a literature review.

There are a few spellings that I would suggest correcting. Please search for and correct British spellings such as: counselling, utilise, analysed, etc. Overall, the writing style is very good.

The paragraph (lines 97-117) is difficult for the reader. Consider presenting this information in a different format that is more reader friendly (e.g., a table format with bullets).

The flowchart (Figure 1) outlines the process well regarding article identification and selection.

A limitation of the literature is that only 11 articles were found eligible for review. I believe this sentence should be added to the limitations section (lines 377-380).

Regarding the conclusions. I like the last statement in the abstract (conclusion) that reads (lines 30-31): “More research is necessary to support pharmacists in this role.” When you read the article, this general statement is missing and the closest sentence to this is (lines 387-389): Government and industry support is required to fund more research into current practices and potential roles for pharmacists in sports medicine.” I suggest incorporating the more general statement (lines 30-31) in the conclusions section (lines 381-389) as well as the sentence (lines 387-389) is too specific and limiting – research in this area could be accomplished outside of the government and industry and may not need funding. Consider the following (to be added to the conclusion paragraph):

“More research is necessary to support pharmacists in this role. For example, government and industry support is needed to fund more research into current practices and potential roles for pharmacists in sports medicine.”

Author Response

(The authors gave the same response as above.)
